# Transcriptome Shock in Developing Embryos of a *Brassica napus* and *Brassica rapa* Hybrid

**DOI:** 10.3390/ijms242216238

**Published:** 2023-11-12

**Authors:** Weixian Zhou, Libin Zhang, Jianjie He, Wang Chen, Feifan Zhao, Chunhua Fu, Maoteng Li

**Affiliations:** 1Department of Biotechnology, College of Life Science and Technology, Huazhong University of Science and Technology, Wuhan 430074, China; m202172236@hust.edu.cn (W.Z.); libinzhang@hust.edu.cn (L.Z.); jianjie_he@hust.edu.cn (J.H.); m202172334@hust.edu.cn (W.C.); d202381023@hust.edu.cn (F.Z.); fuchunhua@mail.hust.edu.cn (C.F.); 2Key Laboratory of Molecular Biophysics of the Ministry of Education, Wuhan 430074, China

**Keywords:** transcriptome shock, expression level dominance, homoeolog expression bias, RNA sequencing

## Abstract

Interspecific crosses that fuse the genomes of two different species may result in overall gene expression changes in the hybrid progeny, called ‘transcriptome shock’. To better understand the expression pattern after genome merging during the early stages of allopolyploid formation, we performed RNA sequencing analysis on developing embryos of *Brassica rapa*, *B. napus*, and their synthesized allotriploid hybrids. Here, we show that the transcriptome shock occurs in the developing seeds of the hybrids. Of the homoeologous gene pairs, 17.1% exhibit expression bias, with an overall expression bias toward *B. rapa*. The expression level dominance also biases toward *B. rapa*, mainly induced by the expression change in homoeologous genes from *B. napus*. Functional enrichment analysis revealed significant differences in differentially expressed genes (DEGs) related to photosynthesis, hormone synthesis, and other pathways. Further study showed that significant changes in the expression levels of the key transcription factors (TFs) could regulate the overall interaction network in the developing embryo, which might be an essential cause of phenotype change. In conclusion, the present results have revealed the global changes in gene expression patterns in developing seeds of the hybrid between *B. rapa* and *B. napus*, and provided novel insights into the occurrence of transcriptome shock for harnessing heterosis.

## 1. Introduction

Whole-genome multiplication usually occurs during the evolution of crops and is accompanied by the production and stable transmission of superior traits [1]. Polyploidy is a significant feature of existing plant diversity [2], and many originated from interspecific hybridization [3]. It was revealed that the heterozygous polyploids frequently exhibited a greater capacity for environmental adaptation [4,5], which could primarily be explained by transcriptional regulation [6]. The phenomenon has also been demonstrated in *Brassica*, and it was found that the transcriptional and splicing changes caused by genome fusion could be recovered to parental levels by genome doubling in allopolyploids [7,8]. However, the exact reasons for this phenomenon are yet to be fully elucidated.

In heterozygous polyploids, the transcriptome shock results from interspecific hybridization rather than genomic polyploidization, usually leading to genomic fusion and introgression from different parents [9,10]. In addition, the duplicated genes with varying levels of expression from the parental additivity could be described as non-additive genes in polyploids [11]. The homologous expression bias (HEB), transgressive expression, and expression level dominance (ELD) were the main manifestations of this phenomenon [12]. HEB refers to the different expression levels of individual homologous genes in the hybrid [13], which mainly reflects the legacy of the progenitors [14]; it is generally measured by comparing differences in the expression of homologous gene pairs in an individual. ELD indicates that the total expression level of a homoeologous pair in hybrids is similar to one parent, but different from the other [15]. It is another kind of crucial homozygous expression bias, which does not consider the relative expression levels of individual homozygotes, but refers to the total expression levels of homozygous gene pairs in heterozygous polyploids compared to their relative expression in both parents [16]. That is, HEB focuses on the relative expression of individual homologous genes. In contrast, ELD focuses on the total expression of homologous gene pairs in the hybrid progeny concerning their expression in both parents. In addition, transgressive expression means that the total gene expression is lower or higher than that of both parents [11]. Previous studies have established that transcriptome shock might arise due to parental influence, gene dosage balance, transcription factor, and cis-/trans-regulatory networks [17,18,19]. The ELD over one of the parents is mainly due to the up- or down-regulation of the homologous genes of the “non-dominant” parents [20]. It is likely to be associated with TFs between progenitors [18]. In this context, it would be interesting to explore whether the expression level of candidate TFs differs between homologous genes. Alongside that, previous studies also found that the gene expression shifts in the polyploid progeny were related to transposable elements [21] and SNPs [22], etc. 

Through epigenetic and multi-omics analyses, transcriptome shock has been widely studied in Arabidopsis thaliana, cotton, wheat, maize, and rice [23,24,25,26,27,28,29,30]. The studies of gene bias were also performed in the genus *Brassica*. For example, in the leaves of *B. napus* (AACC) polyploid, most of the genes were slightly biased toward the A subgenome (36.5%), and low expression of ELD occurred due to the down-regulation of genes from non-dominant parents [31]. Notably, the gene expression bias varies among plant tissues [13,15,16]. Also, in *B. napus*, more gene pairs showed ELD-A in stems and silique, while the opposite was observed in leaves and flowers [15]. Interestingly, gene expression bias might change in different parts, even in the same organs. For example, the gene expression bias in the silique’s upper and lower parts differed in the artificially synthesized intergeneric hybrids of *Raphanobrassica* [16]. However, few studies have investigated the global changes in transcriptional levels in hybrid plants during a very early developing period.

*B. napus* (AACC, 2n = 4x = 38) is one of the most important oilseed crops, which is widely cultivated in China. *B. rapa* (AA, 2n = 2x = 20) is frequently used as an important genetic resource for the genetic improvement of *B. napus* [32,33,34]. In this study, a synthetic hybrid (AAC, 3x = 29) was constructed by hybridization between *B. napus* (maternal parent) and *B. rapa* (parental parent). In order to reveal the underlying basis of gene expression, comparative transcriptomic analyses were performed in developing embryos of *B. napus*, *B. rapa*, and their hybrids, and the gene expression patterns and phenomena of transcriptome shock in hybrid were obtained.

## 2. Results

### 2.1. Phenotypic Observations of the Parents and Their Hybrid

The phenotype of the seeds from self-crossed *B. napus*, self-crossed *B. rapa*, and their reciprocal crossed progenies were first compared. It was shown that within 30 days after pollination, the color of the self-crossed seed coat in *B. rapa* had changed to darker; however, the seed color of the hybrid was greener and lighter, regardless of whether *B. rapa* or *B. napus* was used as the male parent (Figure 1A). The number of seeds in the silique of hybrids was significantly lower than that in the parents. Further analysis showed that the seed number per silique in hybrids was much lower when *B. rapa* was the female parent (6.1 ± 1.8) than when *B. napus* was the female parent (15.5 ± 2.5) (Figure 1B). The seed diameter of the hybrids obtained using *B. rapa* as the male parent (2.67 ± 0.15 mm) was similar to that of *B. rapa* (2.6 ± 0.1 mm), which was bigger than that of *B. napus* (2.3 ± 0.3 mm) (Figure 1B). In comparison, the seeds were much smaller (2.0 ± 0.1 mm) in hybrids when *B. napus* was the male parent (Figure 1B).

### 2.2. Global Gene Expression Analysis of the Parents and Interspecific Hybrids

The synteny analysis between *B. napus* (female parent) and *B. rapa* was first performed. It was shown that the A^n^ subgenome of *B. napus* and A^r^ subgenome of *B. rapa* had high collinearity (Figure 1C), as *B. napus* is generally believed to have originated from a cross between *B. rapa* and *B. oleracea* (CC, 2n = 18) [35]. The transcriptome sequencing was performed in *B. rapa*, *B. napus*, and their hybrids (A^n^A^r^C^n^), and about 25,000,000 clean reads of each sample were obtained after image identification, decontamination, and dejunctioning of the secondary sequencing data (Appendix A). HISAT2 and bowtie2 analyses showed that the average percentage of uniquely matched reads to *B. napus* was about 60~70%, while the number of multiple-matched reads was around 10% (Appendix A). PCA analysis revealed that the samples were grouped into three clusters (Appendix A), and Spearman’s rank correlation analysis showed that the coefficient among biological replicates in each cluster was above 0.85 (Appendix A). To validate the differential expression of genes among these three species, 12 genes were randomly selected for RT-qPCR analysis, and the expression tendency was the same as the transcriptomic sequencing results (Appendix A).

The global expression patterns in the parents and hybrids were analyzed, and a total of 121,237 genes for FPKM were tested. It was found that about 70,000 genes were not expressed (FPKM = 0) or had low expression levels (FPKM < 1) in all three materials (Figure 2A). In hybrids, 33,929 and 14,938 genes were with 1 ≤ FPKM < 10 and 10 ≤ FPKM < 100, respectively, and 11,935 and 6741 were expressed in all three materials (Figure 2B,C). Notably, 2208 genes in the hybrids were extremely highly expressed (FPKM ≥ 100), and 943 of them were constitutively expressed in all three materials (Figure 2D). In short, about 35% to 45% of the genes were all constitutively expressed in the parents and hybrids, respectively. The global analysis of 24,060 genes with FPKM ≥ 2 in all three materials revealed that the overall expression pattern of the hybrids was more similar to that of *B. rapa* than *B. napus* (Figure 2E). Based on the expression analysis results, we performed a differential expression gene analysis (Figure 2G). Compared to *B. rapa*, the up-regulated (Log2FC > 2) and down-regulated (Log2FC < −2) DEGs in *B. napus* were 28,033 and 11,153 genes, respectively. In the hybrids, the 7162 up-regulated and 884 down-regulated genes, 16,724 up-regulated and 18,357 down-regulated genes, were identified compared to *B. rapa* and *B. napus*, respectively. Similar to the previous results (Figure 2E), the highest number of DEGs was found between the two parents. The hybrid had 35,099 DEGs with a stable distribution of up- and down-regulation compared to *B. napus*, but only 8046 DEGs compared to *B. rapa*, and there were much more up-regulated DEGs than down-regulated DEGs. The number of DEGs between the hybrids and *B. rapa* was only a quarter of that between the hybrids and *B. napus* (Figure 2G). This result might be related to the imbalanced genetic information in the hybrids; that is, only one subgenome (A) of the hybrids was from *B. rapa*, and the other two subgenomes (A and C) were from *B. napus*.

### 2.3. Differential Expression Profiling of the Parents and F1 Hybrids

KEGG enrichment was performed to analyze the functions of DEGs (Appendix A). A total of 28 significantly different (*p* < 0.05) KEGG pathways were enriched between the two parents, mainly for unsaturated fatty acid synthesis, photosynthetic carbon fixation, oxidative phosphorylation, TCA cycle, and amino acid synthesis pathways (Appendix A), which indicated that there were differences in the energy metabolism and material synthesis in the embryos between the two parents. Thirty-four and twenty-one different KEGG pathways were enriched in the hybrid compared to the female and male parent, respectively. It showed that the most enriched terms between the hybrids and parents were the genetic information processing and metabolism categories, which suggested that gene information transmission and material synthesis are very active in the embryo. Interestingly, the up-regulated DEGs in hybrids were involved in the oxidative phosphorylation and TCA cycle pathways compared to the *B. rapa*. Moreover, the DEGs involved in carbon fixation and unsaturated fatty acid synthesis pathways were up-regulated in hybrids compared to the *B. napus*. Further analysis showed that DEGs associated with the brassinosteroid synthesis and photosynthesis pathways were up-regulated in the hybrids compared to both parents. Those pathways are necessary for nutrient accumulation and plant growth (Appendix A). GO enrichment analysis revealed that the DEGs between the parents and hybrid were enriched mainly for the biological processes (such as carbon metabolism, apoptosis, and stress resistance) and molecular functions (such as water transport and organic matter transport) (Appendix A). 

The genes associated with embryo development (listed in Appendix A) were further selected for KEGG analysis. It was shown that the environmental adaptation and metabolism were significantly enriched (Appendix A). Take AAC compared to AACC; the DEGs involved in cell division, stalk growth, and cell elongation were up-regulated, consistent with the phenotype presented by the seed (Figure 1 and Appendix A). For DEGs involved in the resistance gene induction pathway, the expression in the hybrid was higher than that of the two parents (Appendix A).

### 2.4. Genome-Wide Unbalanced Biased Expression toward B. rapa in the Hybrids

The biased analysis of homologous gene pairs between the A^r^ and A^n^ genomes and the A^r^ and C^n^ genomes was analyzed using the BBH method to investigate the HEB in AAC embryos. A total of 14,423 and 8907 homologous gene pairs were identified for the A^r^ and A^n^ genomes, and A^r^ and C^n^ genomes, respectively, and were used for HEB analysis. The results revealed that both homologous gene pairs were substantially biased toward A^r^A^r^ (Figure 3A,B). Specifically, among A^r^/A^n^ genomes, there are 20.9% (3012) AA-biased genes and 2% (293) AACC-biased genes. On the other hand, in A^r^/C^n^ genomes, there are 73.8% (6577) AA-biased genes and 0.5% (46) AACC-biased genes. The phenomena imply a more pronounced gene bias of homologous gene pairs within the A^r^/C^n^ subgenomes.

Notably, a more significant proportion of genes biased toward AA were found in the homologs between the A^r^/C^n^ genomes compared to homologs between the A^r^/A^n^ genomes. For example, 77.1% (11,118) and 25.6% (6577) of the homologous gene pairs between the A^r^/A^n^ genomes and A^r^/C^n^ genomes were unbiased, and it seemed that the genomes with a relatively lower homology hold more biased homologous gene pairs (Figure 3). Many biased homologs with opposite expression patterns to the parents emerged, with 9% (1301) in the A^r^/A^n^ genome homologs and 33.1% (2958) in the A^r^/C^n^ genome homologs. In addition, 53.4% (7257) of homologs between the A^r^/A^n^ in the hybrids and 56.6% (5035) homologs between the A^r^/C^n^ in the hybrids held the conditions of the parents, which maintained the stability of gene expression balance.

### 2.5. Genome-Wide Expression Level Dominance Biased toward AA in the Embryo of AAC

To detect additivity, transgressive regulation, and expression level dominance, we classified the homologous pairs obtained by comparing the total expression of homozygous pairs in AAC with that in the parents into 12 categories (Figure 4A). Of the homologous gene pairs between the A^r^ and A^n^ genomes and the A^r^ and C^n^ genomes, 47.1% and 45.0% showed equivalent expression to parents in the hybrid, respectively. Of the homologous pairs between A^r^ and A^n^, 6559 (45.5%) showed ELD and 93% showed ELD of Ar gene expression. Of the homologous pairs of A^r^ and C^n^, 3655 (41%) showed ELD and 72.2% exhibited ELD of A^r^ gene expression. In addition, 3.4% and 8.7% of the A^r^/A^n^ and A^r^/C^n^ homologous pairs showed additive expression. Further analysis revealed that homologous pairs with transgressive up-regulation and transgressive down-regulation were only 4.9% and 4.4%, respectively. Thus, most homologous gene pairs showed dominant and changeless expression in the newly synthesized hybrid.

The expression changes in homologous genes from both parents that appeared ELD were also analyzed to explore the relationship between ELD and individual homoeolog expression levels (Figure 4B). One or both homologs altered their expression after merging the two parental genomes, which led to ELD. Interestingly, in the homologous gene pairs of the A^r^ and A^n^ genomes in hybrids, the reason for the occurrence of A^r^ genome-biased ELD (categories II and XI) is that after the genome merger, 3829 homologous genes from A^n^ were up-regulated, and the expression level of the homologous genes from A^r^ was unchanged. In the homologous gene pairs of A^r^ and C^n^ genomes, it is evident that the ELD in favor of the A^r^ genome (categories II and XI) was caused by 1212 down-regulated homologous genes from the C^n^ genome after genome merger, while the expression level of the homologous gene from the A^r^ genome remained unchanged. In conclusion, ELD could be explained by regulating homologous genes in non-dominant parents. 

### 2.6. The Functional Analysis of ELD Genes

The GO and KEGG enrichment of ELD genes were performed to analyze the functions (Appendix A). It was shown that the genes of A^r^-ELD were enriched for photosynthesis, brassinosteroid, and chloroplast, which indicated that the hybrids were more similar to *B. rapa* in terms of growth and substance accumulation (Appendix A, Table 1). It is worth noting that the transgressive up-regulated genes were mainly enriched in response to the absence of light. Considering that the earlier development of embryos from self-crosses and interspecific crosses was in a bagged environment, the embryos of the hybrids might have a specific development advantage under these conditions.

A^r^-ELD genes were enriched in various KEGG pathways, including chlorophyll, carotenoids, and other phytochromes, which explains the similarity of the seed color of the hybrid to *B. rapa*. In addition, the transgressive down-regulated genes were mainly found in the homologs of A^r^ and C^n^, while transgressive up-regulated genes were found primarily in the homologs of A^r^ and A^n^. The former were enriched to the steroid synthesis pathways, α-linolenic acid metabolism, and flavonoid synthesis. In contrast, the latter were enriched to the plant disease resistance pathway, fructose and mannose metabolism, oxidative phosphorylation, zeatin biosynthesis, etc. This part of genes is theoretically critical in the generation of hybrid advantage, and the results obtained from the enrichment of embryonic transcriptome data showed that the hybrid might have advantages in growth and development, as well as disease resistance.

The hybrids from the hybridization between *B. rapa* and *B. napus* had been reported to be more viable, precocious, and have higher drought tolerance [36,37,38,39]. Brassinolides are critical plant growth regulators and play an essential role in the growth and development of *Brassica* and disease resistance [40,41,42,43]. The expression of key genes in the brassinolide synthesis pathway revealed that the overall gene expression was biased to *B. rapa*, with higher expression of certain genes than both parents (Appendix A). 

### 2.7. Transcription Factor Analysis of Non-Additive Genes

Global transcriptional regulation is closely linked to transcription factors (TFs). The present study predicted 602, 47, and 39 TFs in A^r^-ELD genes, transgressive down-regulation genes, and transgressive up-regulation genes, respectively (Appendix A). Most of these TFs belonged to the MYB family, which regulated the plant hormone signal transduction pathway (Figure 5A,B). We then focused on the TFs of expressed A^r^-ELD genes in the hybrid with higher expression levels than that of female parents (FPKM > 10, log2 fold change >4) (Figure 5C), of which eight TFs were selected. Through exploring their homologous gene functions in *Arabidopsis*, *GeneID:106356676* and *GeneID:106439083* related to cytokinin signaling and physiological processes were reported to be expressed explicitly in the endosperm at early developing stages [40]. In addition, *GeneID:106422025*, *GeneID:106443040*, *GeneID:106362303,* and *GeneID:106432055* are associated with freezing tolerance [44], cytokinin correspondence, development regulation of shoot stem cells [45], seedling photomorphogenesis, and hypocotyl elongation regulation [46], respectively. *GeneID:106369326* was reported to have a critical regulatory function in chloroplast development and embryogenesis [47,48,49]. In addition, the expression level of the TFs in *B. rapa* and hybrid is much greater than that in *B. napus*, which could be an important factor in explaining the differences in embryonic development between the three materials. Subsequently, the TFs and their interacting proteins were further analyzed (Figure 6), and a total of 111 interacting proteins were found. It exhibited that many proteins were enriched in “response to hormones”, “response to stimulus”, “nitrogen compound metabolic processes”, etc. Among them, some proteins were reported to have unique functions, including the *MEE5* gene, which was associated with embryo development and the development of the female gametophyte [50,51]. In addition, *LSM2* [52] and *RAP74* [53] regulate transcript splicing and thus influence plant development, and *STZ* [54] was related to drought stress resistance. These results suggested that the expression bias of the key TFs in the hybrid might be an important factor influencing the global state of transcription and the phenotype.

## 3. Discussion

Interspecific crosses are often used in *Brassica* crops to penetrate superior traits, improve crop yield, confer stress resistance, etc. [55,56,57,58]. The progeny of the crosses between *B. rapa* and *B. napus* has been reported to have advantages such as fast growth rates and high-stress tolerance [35,59]. Because the benefits of heterosis may start very early in crop growth, this study selected early embryos of *B. rapa*, *B. napus*, and the hybrid for transcriptome analysis, and explored the phenomenon of transcriptome shock. The seed maturation rate (seed coat discoloration rate) was higher in *B. rapa* than in *B. napus*. In comparison, the hybrids’ development rate was faster than *B. napus* (Figure 1A). Also, the hybrid’s embryo seems to have more photosynthetic pigments, considering its darker color (Figure 1A). The period of embryo development involves the synthesis and accumulation of various nutrients and significantly impacts the later germination of seeds [60,61]. Transcriptome results showed analogous gene expression patterns in all three species. They found that the DEGs were mainly enriched in “substance synthesis”, “stress resistance”, and “genetic information processing” (Appendix A), which was consistent with the phenotypic difference in developing speed exhibited by the three species. 

In this study, approximately 50,000 expressed genes (FPKM ≥ 1) were detected in the hybrid, and 70% were differentially expressed compared to the female parent. Further analysis revealed that the up-regulated genes were almost equal to the down-regulated genes (Figure 2G). In contrast, only 16% of the expressed genes were differentially expressed compared to the male parent, and the up- and down-regulated genes were highly unbalanced. This phenomenon was similar to the previously reported balanced regulation of gene expression during polyploidy formation in *B. napus* (AACC) [62]. At the same time, some studies have reached the opposite conclusion; for example, it was shown that 85% and 87.5% of DEGs were down-regulated and up-regulated in re-synthesized *B. napus*, respectively [31,63]. Increasing the gene and genome dosage in polyploids usually leads to genome instability, chromosome imbalance, and regulatory incompatibility; reprogramming homologous gene expression is essential for genome and organism stability [17]. Combined with the present and previous studies, this reprogramming of gene expression varies in different parts of the plant and at different developing periods, and may be influenced by environmental factors.

There are many studies on heterosis and homozygous gene bias in the genus of *Brassica*. In *B. napus*, most genes were biased toward the A subgenome [31]. Previous research revealed that the ELDs differed in different tissues in *B. napus*. The gene pairs in ELD-A were more than that of ELD-C in stems (13% ELD-A vs. 10.8% ELD-C) and longhorn fruit (25% ELD-A vs. 17.6% ELD-C); otherwise, the gene pairs in ELD-C were higher than that of ELD-A in leaves (3.6% ELD-A vs. 5.8% ELD-C) and flowers (21.1% ELD-A vs. 22% ELD-C) [18]. The transcriptome shock was examined in the young leaves of eight synthetic *B. napus* plants that were produced from crossing and colchicine doubling, and it was shown that the ELDs in all materials were biased to the A genome, but with significant distinction [64]. Transcriptome analysis of lncRNA from the young leaves of hybrid progeny between *B. napus* and *B. rapa* revealed ELD [65]. In this study, analysis of HEB and ELD revealed that transcriptome shock was indeed present in the developing embryos of the hybrid progeny. 

Approximately 20% of genes homozygous for the A^n^ and A^r^ subgenomes exhibited A^r^-biased HEB, and about 70% of genes homozygous for the C^n^ and A^r^ subgenomes showed A^r^-biased HEB (Figure 3). ELD analysis was consistent, with 42.3% and 29.6% of genes homozygous for the two groups exhibiting A^r^-ELD, respectively. The association of ELD with individual expression dominance analysis revealed that ELD was mainly due to changes in the expression of genes from the *B. napus* subgenome, and the expression levels of genes from *B. rapa* remained stable (Figure 4). Perhaps the *B. rapa* genome regulators tend to regulate gene expression from other genomes after genomic fusion. It is worth noting that the chromosome composition of the heterozygous triploid AAC used in this study is relatively complex. The regulation of the genome itself, dosage effects from different parental genomes, nucleolar dominance hierarchy of different types of chromosomes [17], and deficiencies in statistical methods are all influential factors in the study of transcriptome shock. Developing transcriptome shock analysis methods suitable for species with complex chromosome compositions is one of the following research goals. In summary, in *Brassica*, for different species and tissues, transcriptome shock is common in hybrid plants, and gene expression patterns generally vary, showing a bias toward a particular subgenome. Identifying universal patterns through published studies is complex, and further research is still needed.

For the non-additive genes found above, we explored their functions using KEGG and GO analysis (Table 1 and Appendix A). Since the production of heterosis is usually associated with transgressive regulated genes [66], we focused on the function of this part of the genes. Transgressive up-regulation was mainly found in homologous within the A^r^/A^n^ subgenome (71.4%), while transgressive down-regulation was found primarily in homologous within the A^r^/C^n^ subgenome (88.6%). Notably, the transgressive up-regulation genes were enriched mainly in response to the absence of light, suggesting a development advantage for the hybrid progeny in the extreme lack of light in embryonic development. In the functional analysis of ELD genes, A^r^-ELD was enriched for a range of photosynthesis-related terms, which could explain, to some extent, the superior developing speed of hybrid embryos compared to the *B. napus* phenotype. We also focused on the expression patterns of the genes for brassinolide synthesis, which is closely related to plant growth and development, photosynthesis, and stress resistance [67,68,69,70]. The expression pattern was consistent with the phenotype (Appendix A). The hybridization between *B. napus* and *B. rapa* was the primary method in rapeseed breeding [33], but it was frequently met with seed abortion. The present study found that the genes involved in photosynthesis and the regulation of some growth hormones were up-regulated in triploid hybrids; it indicated that the seed yield might be increased by the overexpression of these genes in future breeding.

There are many potential mechanisms causing transcriptome shock, such as transposable elements [71], alternative splicing [72], and epigenetic modification [73,74], which have yet to be fully elucidated so far. Cis- and trans-regulation are among the most critical causes of transcriptome shock, and transcription factors significantly influence the generation of transcriptome shock [18]. Previous studies have identified more trans-regulated genes than cis-regulated genes in hybrids, possibly due to phylogenetic distance between parents [12]. Therefore, we focused on analyzing the interaction network of transcription factors in non-additive genes (Figure 6). Eight transcription factors with high expression and large differences in expression between progeny and parents were screened within the A^r^-ELD genes. These TFs and their regulated downstream genes had significant roles in plant growth, development, and stress resistance. It is worth noting that changes in variable splicing are an essential component of the transcriptome shock experienced by new heterozygous polyploids [75], and genes such as *LSM2* and *RAP7* in the TF regulatory network can regulate variable splicing of plant transcripts [55,56], which may also be responsible for the transcriptome shock. The next step could be constructing a pooled CRISPR library for the functional validation of these genes, as previously described [76].

## 4. Materials and Methods

### 4.1. Plant Materials

For convenience, we have labeled the species names in the subgenomes in the table below, e.g., A^n^ denotes the A subgenome of *B. napus*. The triploid hybrid (A^n^A^r^C^n^, 3x = 29) was produced by the hybridization between *B. rapa* var. Tianmen Youcaibai (Male parent, A^r^A^r^, 2n = 2x = 20) and *B. napus* var. Huashuang 3 (A^n^A^n^C^n^C^n^, 2n = 2x = 38). The *B. napus* seeds were kindly provided by Prof. Jiangsheng Wu of Huazhong Agricultural University. The embryo materials for the phenotypic observations were taken from the lateral branches with three replicates. Statistical analyses were carried out using GraphPad Prism 9.5. The developing embryos, 21 days after pollination, were taken out and used for RNA sequencing.

### 4.2. Collinearity Analysis

The protein, CDS FASTA files, and GFF files for *B. napus* and *B. rapa* were downloaded from the Ensemble Plants database (https://plants.ensembl.org/, accessed on 15 April 2023). Homologous genes were obtained using BLASTP (blast−2.9.0+) with a cutoff e-value 1 × 10^−10^. Moreover, only the first transcript was used when the gene had more than one transcript. Then, they were used for collinearity analysis using the WGDI [77].

### 4.3. RNA Isolation, Sequencing, and Data Quality Detection

Total RNAs were isolated using a TRIzol reagent (Invitrogen, Carlsbad, CA, USA). RNAs from developing embryos of the plants mentioned above were used to perform high-throughput sequencing on the Illumina HiSeq 2000 platform. After base calling and precision filter, clean data were obtained. HISAT2 was used to map the second-generation sequences of each sample with the reference genome sequence. On the other hand, bowtie2 was used to align the second-generation sequence to the reference transcript sequence after quality control. Then, using RSEM, bowtie2 comparison results were called for statistical analysis. To obtain the number of reads, each sample was compared to each transcript. The FPKM (fragments per kilobase per million bases) conversion was performed on them [78].

PCA analysis was performed on all samples’ gene expression values (FPKM). The correlation of the gene expression levels between the samples (Spearman’s correlation coefficient) was analyzed to test the experiment’s reliability and the reasonableness of sample selection [16].

### 4.4. RT–qPCR Validations

The gene expression analysis in the developing embryos of *B. rapa*, *B. napus*, and the hybrids was performed using the RT-qPCR. The ABI 7300 (ABI) was used for the qPCR experiment with three replications, and *ACTIN* was chosen as a reference gene. The relative expression level was calculated by using the delta–delta threshold cycle (Ct) method. The following reaction conditions were used: 95 °C for 1 min, followed by 95 °C for 10 s, 60 °C for 30 s, and acquisition of fluorescence signals, cycling 40 times [79]. The information on selected genes and primers is listed in Appendix A.

### 4.5. Identification of Differentially Expressed Genes and Their GO and KEGG Enrichment Analysis

Differential expression analysis was performed by using DESeq2 in R v4.1.1. Genes with an adjusted *p*-value < 0.05 found via DESeq2 and a |log2 fold change| ≥ 2 were considered differentially expressed [80]. The gglpot2 and pheatmap R packages were used to visualize data.

GO and KEGG enrichment analyses for the DEGs were conducted using the R package GOplot and the KEGG database (https://www.kegg.jp/, accessed on 27 January 2023). Pathway significant enrichment analysis was performed using the KEGG pathway as the unit, and the hypergeometric test was applied to find out the pathways in which the differential genes were significantly enriched relative to all annotated genes. Pathways with FDR ≤ 0.05 were defined as pathways significantly enriched in differentially expressed genes, and we used the R software and set the parameter -fdr to BH (i.e., using BH correction) for pathway enrichment analysis. The GO enrichment analysis method was hyper-geometric distribution, and like the KEGG enrichment analysis method, we selected GO terms with FDR ≤ 0.05 as significantly enriched GO entries [81]. 

### 4.6. Annotation of Homoeologous Gene Pairs

Although the homoeologous linkages are not always one-to-one [31], we assumed that most of them stayed in a one-to-one relationship in the two sub-genomic pairings (A^n^-A^r^, C^n^-A^r^) for convenience. Utilizing protein sequences downloaded from the Ensemble Plants database (https://plants.ensembl.org/, accessed on 15 April 2023), the bidirectional best BLAST alignment approach was used to derive the homoeologs. With a search threshold 1 × 10^−20^, the BLASTP for each sub-genomic pair (retaining only the best matches) was analyzed.

### 4.7. Analysis of HEB and ELD in the Hybrids

We only focused on the genes expressed in at least one parent for HEB and ELD analysis. The expression levels of each homologous gene pair in the two progenitors (A^rp^ vs. A^np^, A^rp^ vs. C^np^, p refers to parent) and the hybrid (A^rh^ vs. A^nh^, A^rh^ vs. C^nh^, h refers to the hybrid) were compared by using Student’s *t*-test (*p* < 0.05) [78]. For the ELD analyses, the sum expression level of the homoeologous gene pair in hybrid was compared to that of the parents by using Student’s *t*-test (*p* < 0.05) [82], namely, (A^rh^ + A^nh^) vs. A^rp^, (A^rh^ + A^nh^) vs. A^np^, (A^rh^ + C^nh^) vs. A^rp^ and (A^rh^ + C^nh^) vs. C^nh^, and other possible categories that including of ELD. Additivity and transgressive regulation genes were also identified [29]. Subsequently, the expression profiles of the two sets of homologous gene pairs between the hybrid and the parents were compared for any significant differences using Student’s *t*-test (*p* < 0.05), and the results were categorized into nine cases corresponding to the nine columns in Figure 4B.

### 4.8. Prediction and Analysis of Transcription Factors of ELD Genes

Hmmscan was used to identify transcription factors using the existing database classification and rules [83]. The protein interaction network was analyzed using the STRINGS database and then visualized using Cytoscape software.

## 5. Conclusions

In summary, we found that the transcriptomic shock in the developing embryos occurred in the triploid of *B. napus* and *B. rapa*. Approximately 20% of the expressed gene pairs displayed the expression bias with a preference to the A^r^-genome. The functional analysis showed that the bias genes were mainly associated with the photosynthesis and phytohormone response pathways. Our present work provides a valuable reference for studying the transcriptome shock phenomenon and mining potential genes in polyploid species.

## Figures and Tables

**Figure 1 ijms-24-16238-f001:**
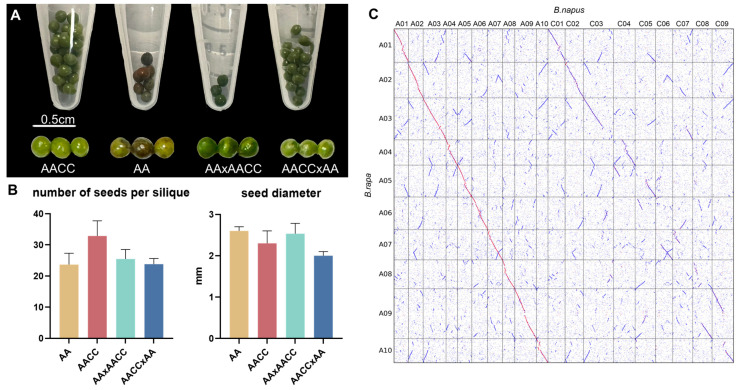
Phenotypes of the parents’ and their hybrids’ seed, and the synteny analysis between *B. napus* and *B. rapa*. (**A**) The seed phenotype comparisons of the hybrid and its parents within 30 days after pollination. Scale bars, 0.5 cm. (**B**) Seed numbers per silique and seed diameter 30 days after pollination. Error bars, standard deviation. (**C**) Dot plot of pairwise synteny between the *B. napus* and *B. rapa* genomes. A01–A10 on the vertical axis indicated the chromosomes of the AA genome. A01–C09 on the horizontal axis showed the chromosomes of the AACC genome.

**Figure 2 ijms-24-16238-f002:**
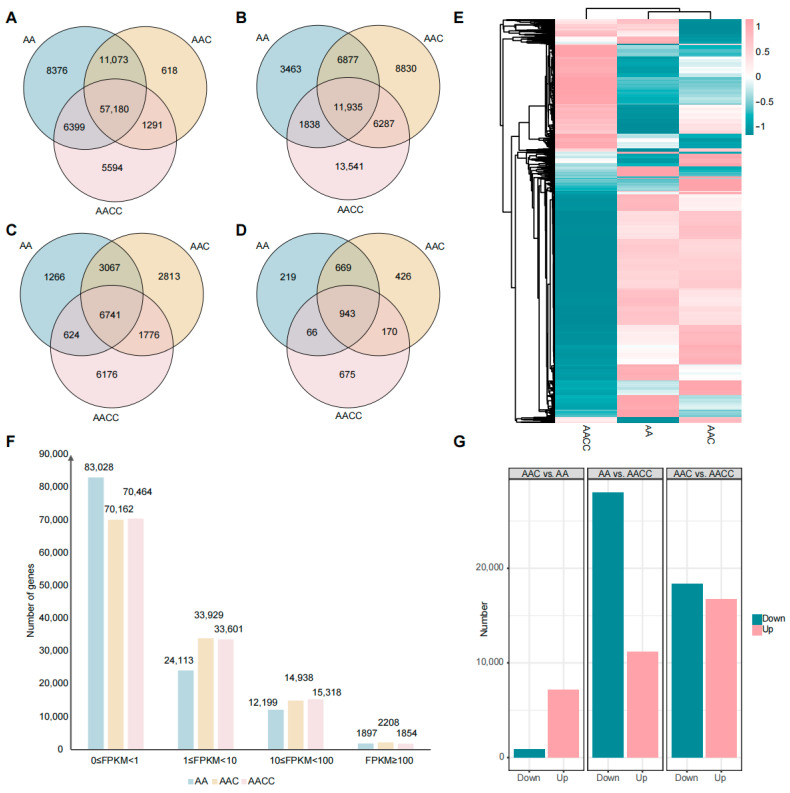
Global analysis of transcriptome data. (**A**–**D**) Venn diagrams of the parents and F1 hybrid with gene expression 0 ≤ FPKM < 1, 1 ≤ FPKM < 10, 10 ≤ FPKM < 100, and FPKM ≥ 100, respectively. (**E**) Heatmap of the expression of 24,060 genes with FPKM ≥ 2 in all three materials using lg(FPKM + 1) as the metric. (**F**) Statistics of the expression patterns in AA, AAC, and AACC characterized by four sets of FPKM values. (**G**) Identification of DEGs in three materials. The ten genes with the most significant expression differences and related information were listed in Appendix A. AAC vs. AACC up-regulation means that the gene is more expressed in AAC than in AACC.

**Figure 3 ijms-24-16238-f003:**
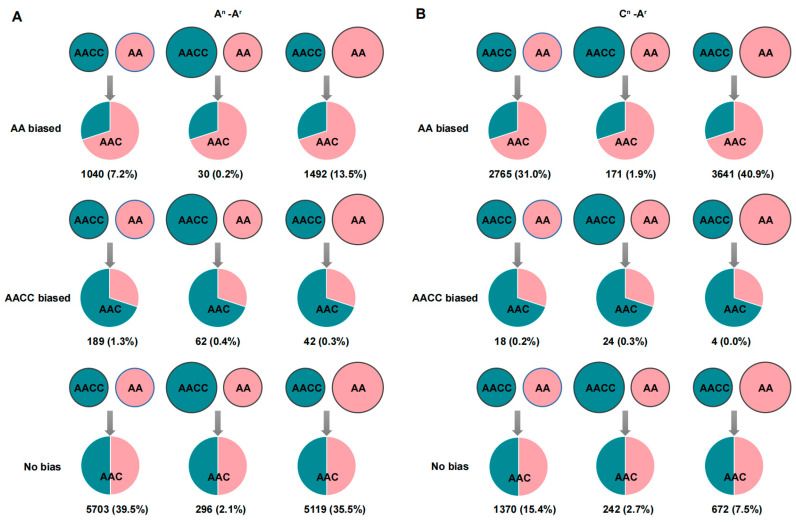
Homoeolog expression bias analysis of homoeologous gene pairs in AAC. (**A**) A^n^-A^r^ genome homoeolog expression bias analyses. (**B**) C^n^-A^r^ genome homoeolog expression bias analyses. The number of homologous gene pairs, as well as their percentage, is indicated in the figure. The size of the circles is only used to distinguish differences in the gene expression of the parents. The ratio of the areas within the circles indicates differences in gene expression levels in hybrids.

**Figure 4 ijms-24-16238-f004:**
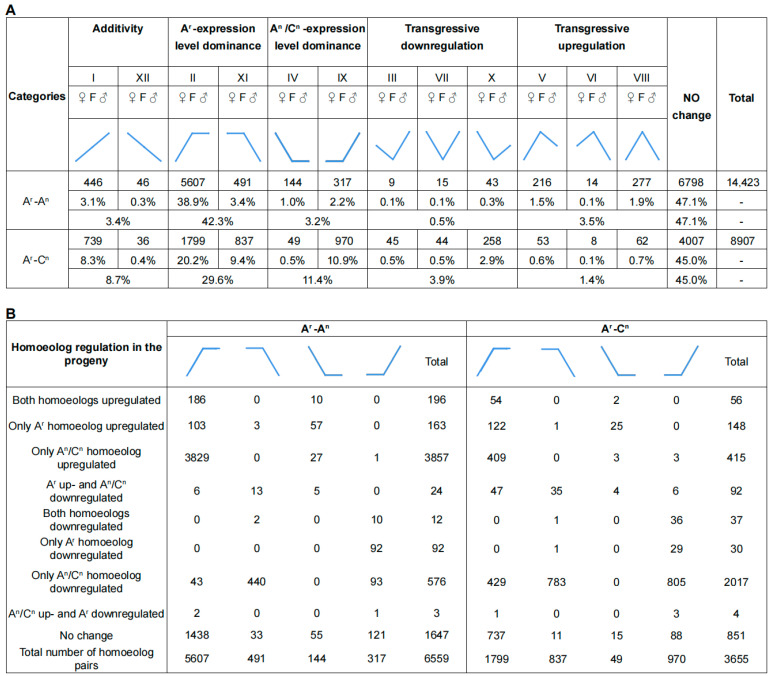
Expression level dominance and relationships between the ELD and individual homoeolog expression levels. (**A**) ELD of the homologous gene pairs in A^r^-A^n^ and A^r^-C^n^. (**B**) Relationships between ELD and individual homoeolog expression levels explained the phenomenon of ELD. The four columns in the figure correspond to the four ELD types of XI, IV, and IX columns corresponding to Figure 4A.

**Figure 5 ijms-24-16238-f005:**
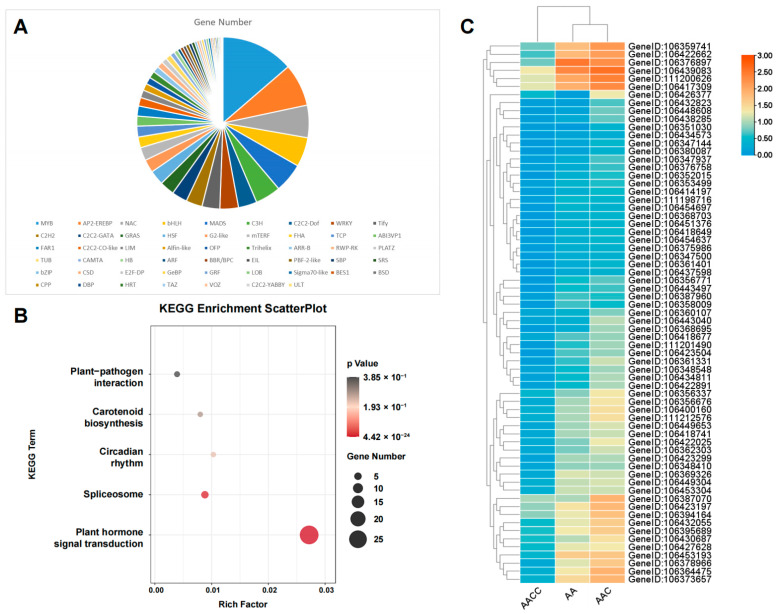
Analysis of the transcription factors in non-additive genes. (**A**) Transcription factor type statistics of A^r^-ELD genes. (**B**) KEGG enrichment analysis of transcription factors. The size of the circle represents the number of genes enriched to this term, and the circle’s color represents the *p*-value. (**C**) Expression profiles of the transcription factors of A^r^-ELD genes (FPKM > 10, log2 fold change >4) in the parents and hybrids.

**Figure 6 ijms-24-16238-f006:**
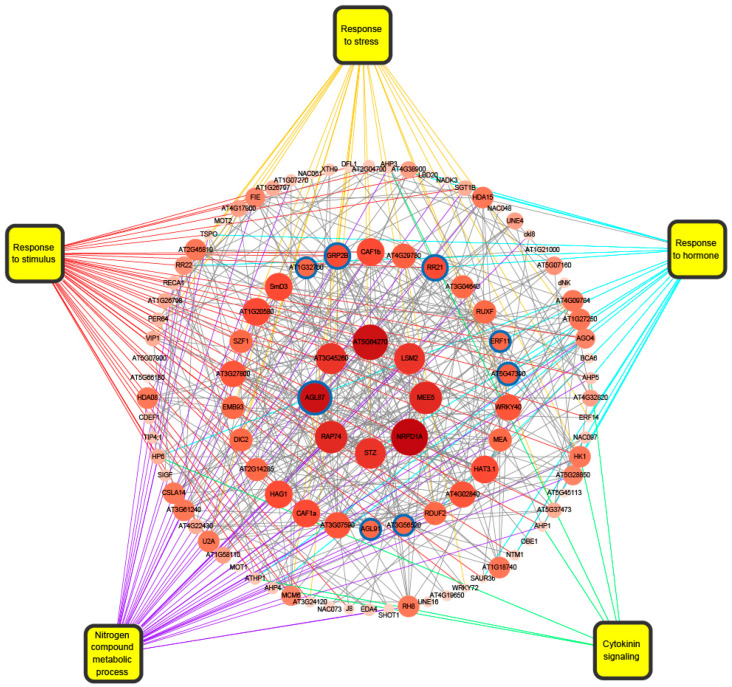
A^r^-ELD transcription factor interactions and function analysis. Blue circles represent target transcription factors. The yellow box represents the comment function terms. Gray lines represent the interactions. Colored lines indicate the functions of the protein.

**Table 1 ijms-24-16238-t001:** Key functions of addictive genes and non-addictive genes.

	Additivity	A^r^-Expression Level Dominance	A^n^/C^n^-Expression Level Dominance	Transgressive Up-Regulation or Transgressive Down-Regulation
A^r^-A^n^	brassinosteroid mediated signaling pathway	translation	DNA replication	response to absence of light
	translation	ubiquitin-dependent protein catabolic process	cellular response to water deprivation	polysaccharide biosynthetic process
	Golgi apparatus	chloroplast	cellular response to salt stress	regulation of plant organ formation
	mitochondrial matrix	mitochondrial matrix	lignan biosynthetic process	drug transport
	origin recognition complex	GTP binding	plant seed peroxidase activity	regulation of flower development
	catalytic activity	ligase activity	transferase activity	alcohol dehydrogenase (NAD) activity
A^r^-C^n^	translation	photosynthesis	small GTPase mediated signal transduction	cellular amino acid biosynthetic process
	cellular amino acid biosynthetic process	phosphorylation	DNA replication	methionine biosynthetic process
	Golgi apparatus	negative regulation of cell death	protein transport	glycogen metabolic process
	cytosol	brassinosteroid homeostasis	DNA replication initiation	microtubule-based process
	RNA binding	chloroplast	protein folding	chloroplast stroma
	endopeptidase activity	ATP binding	chloroplast	plant-type cell wall

## Data Availability

The sequencing raw reads generated in this study were submitted to the NCBI SRA database (https://www.ncbi.nlm.nih.gov/sra/) under accession number SRR26117317, SRR26117316 and SRR26117315 on 10 November 2023.

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
