# Peer review of "Transcriptome Shock in Developing Embryos of a Brassica napus and Brassica rapa Hybrid"

_ijms, 2023, doi:10.3390/ijms242216238_

Round 1

Reviewer 1 Report

Comments and Suggestions for Authors

The study provided insights into the transcriptomic changes due to interspecific crosses in Brassica, contributing to our understanding of polyploid heterosis and the transcriptome shock phenomenon, and offering a basis for future research into the genetic regulation of these processes.

Some comments for the authors to consider:

- It would be beneficial to provide clearer definitions or explanations for specialized terms such as "transcriptome shock," "HEB," and "ELD," which could help readers unfamiliar with these concepts.

- Given the heavy reliance of the paper on bioinformatic tools, it is crucial to offer more detailed descriptions of the methods employed to ensure clarity and replicability.

- The figure legends should be expanded to comprehensively describe the results independently of the main text, thereby enhancing their explanatory power.

- Additionally, the discussion section appears relatively brief. The authors should consider expanding it, for example deeper exploration of the mechanisms behind the observed transcriptomic shock; Potential drawbacks or limitations of the hybridization process; Comparing their findings with those of other studies; considering alternative explanations for the observed results might offer a broader perspective; any practical applications of these findings, such as their impact on breeding practices etc…

Comments on the Quality of English Language

Certain word choices and grammatical structures could be altered in the main text

Reviewer 2 Report

Comments and Suggestions for Authors

Dear Authors,

I have read your paper entitled "Transcriptome Shock in Developing Embryos of Synthesized Hybrid Brassica napus and Brassica rapa" with great interest, which undoubtedly required significant effort on the part of the authors. Nonetheless, I have a few comments to make.

Detailed Comments:

1.     The Abstract is poorly structured. It should include all the points required in the Author Guidelines.

2. In the Introduction, the study states that transcriptomic analyses were conducted on developing embryos of B. napus (AACC, 2n=4x=38, maternal parent), B. rapa (AA, 2n=2x=20, parental), and their hybrids (AAC, 3x=29). Expression patterns of genes were analysed in different materials, and transcriptome shock in the hybrid genome was accordingly revealed. However, there is no clear research objective or alternative research hypothesis stated, in relation to the null hypothesis.

3.     RESULTS. A global analysis of over 24,000 genes with FPKM≥2 in all three materials revealed that the overall expression pattern of hybrids was more similar to B. rapa than to B. napus. Based on the expression analysis results, a differential gene expression analysis was performed. Compared to B. rapa, 28,033 upregulated genes (Log2FC > 2) and 11,153 downregulated genes (Log2FC < -2) were identified in B. napus. In hybrids, 7,162 upregulated genes and 884 downregulated genes were identified compared to B. rapa, as well as 16,724 upregulated genes and 18,357 downregulated genes compared to B. napus. Conclusion: These studies showed that the fusion of two species leads to a transcriptome shock in developing embryos of the offspring, and the analysis of gene functions related to this phenomenon may help understand the mechanisms of heterosis and phenotypic differences between species. This work provides a valuable reference for studies on polyploid heterosis and the understanding of the transcriptome shock phenomenon. In summary, the text is well-written and provides important information about interspecific hybridization and differential gene expression analysis. However, it could be further improved by adding more details about the results and their interpretation.

4. The Discussion contains significant information about the results of interspecific hybridization research in Brassica. The authors focus on differences in gene expression and the mechanisms that influence these differences. The discussion is well-structured and understandable, but it could be more detailed, especially concerning specific results and implications for research on heterosis and Brassica plant development.

5.  There is no Conclusion section, although in the Discussion, the authors formulate a certain conclusion. The conclusion from the discussion is that the fusion of two species leads to a "transcriptome shock" in developing offspring embryos. The authors emphasize the importance of transcriptional regulatory factors in this process and potential mechanisms affecting the "transcriptome shock."

6.     Materials and Methods. The text of this chapter is clear and well-organized, but it requires some comments:

a) Data Sources: You have described the data sources (Ensemble Plants), as well as the tools and methods used for collinearity analysis. This is important for understanding what data were used in the study.

b) RNA Sequencing: The description of the RNA isolation and sequencing procedure is sufficiently detailed. However, it would be worthwhile to provide more information about sequence quality, such as Q-scores.

c) RT-qPCR Validations: The description of the RT-qPCR procedure is clear, and the choice of ACTIN RNA as the reference gene is justified. However, more information about reaction conditions and primers used would be beneficial.

d) Identification of Differentially Expressed Genes: The tools used (DESeq2) and the criteria for differentially expressed genes are clearly presented. This is important for the replicability of the study.

e) Annotation of Homologous Gene Pairs: The description of the BLASTP analysis is understandable, but it would be valuable to specify which database was used for annotation.

I hope that these comments will be helpful in improving your paper.

Sincerely

Comments on the Quality of English Language

Minor English editing is required.
